# Bushfire Smoke and Children’s Health—Exploring a Communication Gap

**DOI:** 10.3390/ijerph191912436

**Published:** 2022-09-29

**Authors:** Erin I. Walsh, Ginny Sargent, Burcu Cevik-Compiegne, Michelle Roberts, Nicola Palfrey, Laura Gooyers-Bourke, Sotiris Vardoulakis, Karima Laachir

**Affiliations:** 1Population Health Exchange, National Centre for Epidemiology and Population Health, Australian National University, Canberra, ACT 2601, Australia; 2Centre for Arab and Islamic Studies, Australian National University, Canberra, ACT 2601, Australia; 3Medical School, Australian National University, Canberra, ACT 2601, Australia; 4Headspace Australia, 1/1 Torrens St. Braddon, Canberra, ACT 2612, Australia; 5Australian Child and Adolescent Trauma, Loss & Grief Network, Australian National University, Canberra, ACT 2601, Australia; 6Healthy Environments and Lives (HEAL) National Research Network; 7National Centre for Epidemiology and Population Health, Australian National University, Canberra, ACT 2601, Australia

**Keywords:** health literacy, wildfires, air pollution, communication

## Abstract

The “Black Summer” bushfires of 2019/2020 in Australia generated smoke that persisted for over three months, mainly affecting Eastern Australia. Most communication strategies focused on the fire itself, revealing a knowledge gap in effective communication of the impact of bushfire smoke on health, especially for children and those living in non-English speaking minority groups. To address this, semi-structured qualitative interviews were undertaken with sixteen adults with caring (*n* = 11) or educational (*n* = 5) responsibilities for primary-school aged children (5–12 years, with some also having children up to 16 years) who had direct experience of the “Black Summer” bushfires. Overall, 43% (*n* = 7) of the sample spoke English as a first language, 25% (*n* = 4) spoke Turkish, with the remainder speaking Persian, Arabic, and Spanish. Thematic inductive qualitative content analysis revealed predominant themes of the role of parents and caregivers as conduits and curators of information. Air quality apps were the most common source of information. Language barriers and the lack of child-friendly methods of communication were highlighted as particular challenges. This qualitative study provides evidence for future development of communication strategies to better serve culturally and linguistically diverse individuals and the children in their care.

## 1. Introduction

Large scale fires (also referred to as bushfires and wildfires) are increasingly impacting human, ecosystem and planetary health [1]. Bushfires are the most common and destructive natural hazard in Australia [2], and profoundly impact human health both directly and indirectly via population exposure to bushfire smoke. Bushfire smoke (especially fine particulate matter) is associated with increases in emergency ambulance dispatches, hospital admissions (both non-emergency and emergency), and increased incidence of chronic obstructive pulmonary disease, asthma complications, and cardiovascular impacts, such as increasing hospitalizations for ischemic heart disease [3]. The impact of bushfire smoke on mental health is less well understood, but experiencing bushfires is associated with elevations in depression, anxiety, and post-traumatic stress [4,5,6].

“Black Summer” was a season of extensive and intense bushfires experienced in Eastern Australia in late 2019 and early 2020 which directly or indirectly impacted over 10 million people [7]. Though it was not the largest or most deadly bushfire in living memory in Australia, it had an extraordinary presence in news and social media [8], and led to unprecedented air pollution levels that persisted for over three months in some of Eastern Australia’s most densely populated areas [9,10,11]. At a peak on January 14th 2020, the population-weighted 24 h PM_2.5_ exposure was 98.5 µg/m^3^, over fourteen times the historical average. Bushfire smoke exposure accounted for 417 excess deaths and 4456 additional hospitalizations nationally [7]. PM_2.5_ and ground-level ozone (O_3_) exposure accounted for an estimated 250 excess deaths and 3490 hospitalizations in New South Wales and Victoria [11]. There were also substantial mental health impacts associated specifically with the smoke, including anxiety, depression, stress, and post-traumatic distress [12,13]. These health and wellbeing effects were magnified in children, especially those in vulnerable communities [14,15].

Through sufficient disaster preparation and response, including prompt and efficient evidence-based communication, harms caused by bushfire smoke can be reduced. However, current practice and translation research to support such communication is underdeveloped. The lengthy fire season put a strain on protocols for acute hazard communication and the ensuing health messaging around bushfires [16]. Given the extreme and prolonged reduction in air quality due to bushfire smoke, existing public health messaging regarding the health impacts of bushfire smoke were insufficient. They focused on short- rather than long-term exposure [17], necessitating ad hoc translation of evidence into more nuanced communication for the general public and at-risk population groups. This generated an additional need from the public for information about how to protect themselves from the impact of ongoing bushfire smoke. A recent review of bushfire smoke exposure communication revealed few strategies are specifically designed for potentially vulnerable populations, including culturally and linguistically diverse communities, whose health is particularly at risk due to socioeconomic marginalization and/or English language limitations [18]. Similarly, ongoing commentary notes how children are at higher health risk, but their perspective is often missing from conversations regarding health messaging around bushfire smoke [19].

The promotion of preparedness and resilience in children is an important component of protecting their capacity to function and adapt during and after an environmental disaster [20]. More broadly, individual decision making is a fundamental building block of population health [21], and relies on health literacy. Health literacy in children regards their ability to meaningfully interrogate and act upon health information, and be empowered with age-appropriate skills and knowledge to engage in autonomous decision making to protect their own health [22,23]. As Paakkari and Okan [24] note, health literacy should be viewed in terms of social responsibility and solidarity. In the case of children, particularly those from culturally and linguistically diverse (CALD) communities, we hold that a child’s health literacy is bound closely to that of their parents, caregivers, and educators. Indeed, the need to include families, schools, and the wider community in considering communications and interventions for children impacted by bushfires was noted by [14,25]. Given the increasing trend of bushfires and associated smoke exposure [26,27], it is timely to understand how we can support health literacy regarding bushfire smoke in children and their support systems.

This study aimed to provide evidence for future development of communication strategies by qualitatively exploring the experiences and communication needs of parents and caregivers of primary-school children during prolonged bushfire smoke exposure, with a focus on perspectives from non-English speaking minority groups. Specifically, the goal was to articulate and make explicit the lessons caregivers learned from their lived experiences of observing and meeting the communication needs of children in their care.

Our conception of “communication” focusses on adult/child information flow regarding the topic of bushfire smoke, and includes both informational and affective content (e.g., the child’s cognitive capacity to understand, and their emotional capacity to cope). We focus on lack of age-appropriate informational resources as the “gap” in this communication, viewed from adult’s lived experience of navigating the needs of children.

## 2. Materials and Methods

The sampling frame was adults with caring and/or educational responsibilities for primary-school aged children (5–12, though some also had children up to age 16) who had direct experience of the 2019 Bushfire season in the Australian Capital Territory and New South Wales (specifically regions close to the Australian Capital Territory such as Queanbeyan). The estimated total population of this region in 2019 was 478,500 (55,500 of whom were children aged 5–12). Given the focus on CALD perspectives, the convenience sample purposively consisted of 23 English, Arabic, and Turkish-speaking adults. The final sample size consisted of 16 individuals (70% response rate). Overall, 43% (*n* = 7) of the sample spoke English as a first language, 25% (*n* = 4) spoke Turkish, with the remainder speaking Persian (*n* = 3), Arabic (*n* = 1), and Spanish (*n* = 1). The sample consisted of four educators, one educator and parent, and eleven parents. Educators worked in primary schools with experience of ages 5–12. Most of the parents/caregivers had two children (range none to three children), with ages ranging between 1 and 14. The ethical aspects of this research was approved by the Australian National University Human Research Ethics Committee. Informed consent was obtained from all participants.

Interviews took place via teleconference in August through December 2020. Depending on participant preference, these interviews were either recorded and transcribed (via otter.io) for analysis, or written notes were taken during the interview. Most interviews (87%) were carried out in English. Nine interviews were recorded and transcribed for analysis, while analysis was based on interviewer notes for seven.

Interviews were semi-structured around the core guiding question “How did you communicate with your children about the need to stay out of the thick bushfire smoke?”

Qualitative analysis built on discussion of recurring themes and concepts emerging from interviews amongst by all authors. Thematic inductive qualitative content analysis [28] was undertaken by one author not involved in interviews based on interviewer notes and interview transcripts (where available in English). Briefly, this technique involves a preparation phase (where sense is made of the interview data as a whole), an organizing phase (where recurrent codes are identified, organized, categorized and abstracted), and reporting of the resultant conceptual map. 

## 3. Results

Figure 1 presents an overview of the inductive qualitative content analysis results.

### 3.1. About the Child

Participants indicated children had varying reactions to the smoke.


*“Every kid has a different experience”—F [educator of children aged 9–12 years]*


The most common impacts of smoke on the child’s daily life reported were having to stay indoors, school being closed, being unable to visit friends or family members, and needing to wear a mask when outside. There was substantial variety in children’s reactions to these directives, and the smoke itself. Some children were content remaining indoors and occupying themselves with pastimes such as video games, while others were discontent with being restricted from outdoor activities such as soccer practice. Levels of anxiety varied, but common themes in the focus of anxiety emerged as worries about the bushfire itself, and concern about the health of others, most notably friends and animals. One educator noted that younger children tended to process their experiences with less factual understanding, and more imagination—*it’s like night outside; it’s scary like a ghost/monster*.

The most common themes in questions posed by children regarded queries about the future duration and severity of the smoke (“How long will the smoke last for?” “When will school start again?”), and perceived danger. Most of the questions about danger focussed on the proximity and risk posed by the bushfire’s flames, rather than the health risks posed by smoke exposure. Several parents noted their children struggled to understand that smoke would impact their health, given benign prior experiences of smoke such as of campfires, while one educator encountered the logic “*It doesn’t smell bad… Why is this dangerous?*

### 3.2. The Child’s Support System

Most respondents experienced considerable distress arising from their experience of the bushfire and accompanying smoke, necessitating composed curation of information they presented to participants. Several parents found it challenging how to speak with children about the broader context of the bushfire smoke (e.g., climate change) without communicating hopelessness or despair. Similarly, several found it challenging to buffer the child from their elderly family member’s distress. Yet, they reflected on children’s need to see the adults around them as calm and in control, feeling it was important to manage their own anxiety in their manner and words around the children, even when they were not directly discussing the bushfire smoke. 


*“It is for the adults to worry”—IS [educator of children aged 8–10 years]*


Many respondents noted that the uncertainty in both the duration and severity of smoke exposure were considerable sources of distress for children. Parents tended to manage this through providing actionable health advice (such as talking about how it is safe inside), while educators found it useful to emphasize to the competence of emergency services, and engage in discussions about how fire fighters are keeping the fires away, and parents and doctors know how to keep them safe from the smoke. 

### 3.3. Information

The most common sources of information regarding the status and spread of bushfire smoke were smartphone apps (Fires Near Me [29]; Air Rater [30]), social media, and mainstream media (television and radio). Several respondents felt a need for frequent updates, so preferred sources such as apps which were regularly updated in near real-time. One respondent reported feeling overwhelmed by the volume of information on social media. When it came to information about the health impacts of the bushfire smoke, source credibility was seen as important, with advice from the World Health Organization being particularly trusted. 

Turning to sources of information directly suitable for children, four participants reported that they felt the television and/or radio news or social media were not suitable sources of information for their children to engage with directly, though some found Internet video sites such as YouTube useful sources of tutorials to show practical smoke protective steps. Those who used air quality apps commonly shared this information directly with their children. They reflected the simple language and colour-coding used by most apps were age-appropriate even for very young children, and helped them understand how the changing situation related to parental instructions (e.g., when it was or was not safe to go outside).


*“They [other parents] were kind of obsessively monitoring the air quality and showing it to their kids everyday because their kids just would not take no for an answer about why they couldn’t go out.”—TC [parent of children aged 5 and 7]*


Most participants expressed that information presented in a manner specifically tailored for children would have been extremely useful to help them communicate more effectively.

### 3.4. Communication Strategies

Risk management and communication was focused on achieving compliance for health-protective behaviours, most notably remaining inside. This was most successfully achieved by framing the adult’s authority as a source of protection, rather than restriction.


*“We need to explain to them that we are keeping them safe. Staying away from the smoke is protecting their bodies.”—C [educator of children aged 8–10 years]*


Education (that is, conveying information particularly around the health risks of smoke) was most successful when description of the physical risks posed by the bushfire smoke were framed in an empowering way, where children had agency to protect themselves. The two facts most commonly raised by respondents were that the bushfire was far away even if the smoke was thick, and that breathing in the smoke is bad for people’s health. No respondent brought to the discussion specific air pollution indicators (such as PM_2.5_ exposure), instead most discussed the impact of smoke in terms of bodily experience, such as itchy eyes. Several parents paired this information with actions the child could take to protect themselves, most notably preventing smoke from entering the home. 

Compassionate communication focused on acknowledging the child’s distress, and reassuring them of the strength and competence of their support system (with a notable recurrent reference to fire fighters keeping the fire away). This was most successful when coupled with transparency—educators in particular emphasized how it was important to be truthful and not keep anything from the children, allowing them to trust in reassurances from adults.

### 3.5. Considerations of Cultural and Linguistic Diversity

Several migrant respondents reflected on having experience of natural disasters, but nothing like Australian bushfires. They expressed that official advice in some cases did not take into account the precarity of their situation—that they or their friends were living in circumstances where smoke may not be avoided, unable to afford air filters, and did not have access to family or friends elsewhere in Australia that could temporarily house them and their children. Notably, several also worried what might happen if their official documents (such as passports and birth certificates) were destroyed, and were unsure of local evacuation protocols. It was difficult to address these ‘bigger picture’ anxieties while meeting the communication needs of their children, who were more focused on the proximal impact of the smoke. A further concern was the need to support other members of their community with low English literacy. Two respondents described a situation where they became de-facto translators for their peers (and, by extension, their children), taking information from television and the Internet and interpreting and distributing it via social media. This highlights the fact that within our sample, and in their communities more broadly, English language proficiency varied substantially. 

Multiple respondents indicated it would be highly beneficial to have resources in their first language to assist them in communicating about bushfire smoke with their children, and other vulnerable members of their communities. They felt this information should include pragmatic, specific guidance that reflected the changing circumstances of the bushfire spread and smoke plume status.


*“You’d be amazed at what the language barrier does.”—SG [parent of child aged 9, Farsi as first language]*



*“So I assume for people that don’t speak English […] Even the elderly people coming from Europe that don’t speak English. We have a lot of Indians in this community. Elderly people, they don’t speak English as well. So I don’t think it was easy for them. They, I’m sure they were looking for someone to translate and keep updating them about the situation day by day.”—TS [parent of children aged 9 and 13, Arabic as first language]*


The language barrier was more than an issue of comprehension. Even if all parties are proficient in English as their second language, they may prefer to communicate in a different language in family settings. In this case, it is more natural and reassuring to communicate health information in that language. Information presented in languages other than English was therefore useful even for those parents and carers who could understand information presented in English.

### 3.6. After the Smoke Clears

In terms of perceived impacts, several respondents reported persistent physical and mental health impacts of the child’s experience of bushfire smoke. Some of those children with perceived health effects had persistent symptoms after the smoke cleared, most notable in those children with asthma. The most common mental health impact was persistent anxiety that the smoke would return, often triggered by ambiguous stimulus in the environment such as cloudy skies, fog, or woodfire heating, or changes in their parent’s behaviour the child associated with the smoke, such as witnessing their volunteer fire fighter father donning his uniform. One parent noted how temperament seemed to be key to a child’s return to normal—even when provided with the same information, one child may experience persistent anxious thoughts while another seems to be content. Positive persistent effects were also reported: one educator (of years 5 and 6) reflected on seeing greater sense of agency and resilience, and a desire to help the environment recover (e.g., organising a market day to raise money for bushfire-affected animals) and addressing underlying causes (e.g., writing letters to their local ministers regarding climate change). 

Five parents reflected on what they had learned, should a similar smoke scenario occur. Most felt they and their children were better equipped to deal with the consequences of the smoke, and have conversations about it’s impact on their lives and health. This was due to feeling they knew more about the topic, and learning from what they saw as mistakes


*“I remember when we were preparing our [emergency] kit, we were hiding it from the children, because we didn’t want them to worry about anything. In hindsight, maybe it was wrong. Maybe I shouldn’t have, you know, as a mother, I was just being protective”—SG [parent of child aged 9, Farsi as first language]*


## 4. Discussion

We have highlighted a number of considerations for the future development of communication strategies to support children affected by chronic exposure to bushfire smoke. Reflections on the “Black Summer” bushfires of 2019–2022 from sixteen educators and parents of primary-school aged children covered the broad themes of the child, the child’s support system, information, communication strategies, considerations of cultural and linguistic diversity, and what happens after the smoke clears.

There were some frequent threads in children’s reactions to bushfire smoke, and their communication needs. The most common experience reported was disruptions to daily life, most notably interruptions in schooling and socialization outside, similar to those experienced by children affected by other bushfires [31,32]. There were also recurrent themes in questions children posed about bushfire smoke, which indicated some degree of agentic health literacy in that they interrogated and acted upon the health information adults provided. There was a notable focus on smoke as a signal of the danger of fire rather than a health hazard within itself, which aligns with evidence elsewhere that children in bushfire situations tend to focus on physical fire proximity rather than other relevant risks [33]. It is possible the immediate destructive consequences of fire are particularly salient to children; psychological distress in children exposed to bushfires is higher if they experience resource loss due to the fire [34].

Though there were some commonality in experiences, there was also substantial variability. Some children took comfort in active information seeking about air quality levels, while others engaged in distracting activities such as playing computer games, a coping mechanism noted in children experiencing a prolonged mine fire in Victoria, Australia [30]. Confusion and anxiety are common in children’s experience of disasters [35], which was raised to some degree by caregivers and educators. However, the severity of perceived anxiety was substantially different across children, as may be expected given evidence for the general principle that children’s responses to high levels of stress are extremely diverse, ranging from mild and short-lived to severe and persistent distress [36]. Two characteristics of a child that can contribute to their experience is mental health before experiencing a disaster, and developmental level [35]. In the current study, educators in particular noted the role of age in children’s recognition, understanding, and reaction to the threat posed by bushfire smoke. Younger children tended to use imagination while older children sought information to guide their behaviour. This reflects Holt et al. [37]’s insight that a child’s level of social-cognitive development influences both how they appraise a threat or source of distress, and how they cope with it, and emphasizes the need to consider specific age and/or developmental level when designing communications intended specifically for children.

Children are part of a family system, and the reaction of that system in its entirety to disasters is highly salient to the child’s experience [35]. A child’s distress is mediated by the reaction of significant adults around them [31], to the point that a parent’s reaction to a traumatic event can be more important than the nature of the event itself (Kinston and Rosser, 1974). Our respondents were aware of this, many reflecting on the importance of projecting a sense of calm in general, and especially when discussing the bushfire smoke. As in the literature surrounding chronic health conditions, uncertainty of outcomes was a major stressor for both the child and their parents, which some successfully managed by focusing on problem-solving [38]. Another source of calm during uncertainty was the presence and competence of fire fighters and related civic support structures. This is congruent with findings elsewhere that the ability of the community to offer support is a key predictor of mental health outcomes in children experiencing natural disasters [35]. 

The breadth of information sources used by our respondents align with information seeking patterns noted elsewhere, e.g., qualitative interviews in 2009 Walla Walla tip fire in [39]. Although respondents in our study relied more heavily on smartphone apps such as Fires Near Me and AirRater than reported in studies relating to earlier fires, our respondent’s use of apps was very similar to findings from a large-scale self-report study regarding AirRater app usage during the Black Summer fires [40]. This may indicate that smartphone air quality apps are gaining prominence as an information source over time. 

We found information needs of adults and children were overlapping but quite distinct, and it is part of an adult’s role to recognize this distinction when communicating with children [25]. Adults required more complete information regardless of how distressing they found it in order to guide decision making for their whole families, while children required sufficient information to guide their behaviour and expectations on a day-to-day basis. The advice obtained and conveyed to children aligned public health advice discussed in other qualitative studies of the Black Summer fires (e.g., [41]).

None of the respondents indicated they had access to information about bushfire smoke intended for children, requiring caregivers and educators to learn from a variety of sources and then tailor and curate the information for children. This can be problematic because it is difficult to tailor health information when you are simultaneously developing your own knowledge on a topic [42]. Several caregivers reflected on sharing smoke monitoring app information with their children, as the frequent updates met the children’s need to understand how their immediate future might be impacted by the smoke, and most of the major apps used motifs such as colour schemes that children could understand. 

We identified three complimentary communication strategies used by caregivers and educators: risk management, education paired with action, and compassion. Taken together, these communication strategies reflect an authoritative approach, where caregiver or educator authority is applied in parallel with age-appropriate explanations of the reasoning underpinning specific requests. Such an approach, rather than permissive or authoritarian approach to parenting, has been associated with greater resilience in adolescents who had experienced a natural disaster [43]. The efficacy of pairing information with corresponding proactive actions that a child could take to protect themselves and others aligns with a larger literature indicating that children in natural disaster settings benefit from feeling like their understanding of the situation renders them competent, helpful actors [32], as productive behavioural responses are particularly helpful for coping with distress [37]. Interviews with children aged 8–11 with experience of the Black Saturday bushfires in Victoria revealed remarkable practicality in their knowledge about what to do [33], indicating this education/action pairing is used by parents and caregivers beyond the current sample. Compassionate communication consisted of acknowledging distress and providing honest reassurance, an approach generally recommended when discussing potentially distressing topics with minors [44,45]. The practice of reminding children of the presence and role of civic supports such as firefighters aligns with findings elsewhere that such an understanding is an important component of children’s capacity to act appropriately during bushfires [46].

We did not find any cultural variation in child’s response or information needs relating to bushfire smoke, though it has been noted elsewhere [35]. This may be due to the limited number of CALD participants in the current sample. Respondents who spoke English as a second language raised the precarity of living in a foreign country, surrounding social support and potential loss of official documents, made their experience of the bushfire and bushfire smoke more stressful, consistent with findings in Howard et al. [47]. The language barrier formed a powerful disruption for adults seeking information to tailor appropriately for their children. While respondents did not reflect more deeply into the nature of the language barrier, research elsewhere indicates the impediment is more than simple understanding of words; it is a mismatch in hearing, understanding, believing, processing, and responding [48]. Notably, who speak English as a second language are faced with the dual task of both translating and adapting information to a level appropriate for their children. Relating this to the language brokerage literature (which typically focusses on children translating for their parents), this can be framed as a task of mediating information rather than transmission, which is a challenging [49]. Parents who spoke English as a second language reported supporting not only their own child, but also their extended same-language community including other people’s children. This highlights the importance of culturally and linguistically diverse social networks for mutual connection, and suggests future disaster preparedness and communication strategies should recognize and leverage these natural social groupings in order to maximize the reach of important information, as advocated in Satizábal et al. [50].

The impacts of Black Summer remained even when the smoke had cleared. This is to be expected, as experiencing a bushfire is potentially a life-changing event that can reframe what is considered ‘normal’ in both parents and children [25]. Educators noted that the experience of bushfire smoke fostered altruism, resilience, and positive action regarding climate change mitigation in some children. This aligns with findings in [51], who further noted an age-dependent effect: after experiencing an earthquake, six-year-old children tended to became more selfish, while nine-year-old children tended to become more altruistic. Another positive reflection was that parents and caregivers both felt they had learned a lot about the health impacts of bushfire smoke, and how to communicate effectively with children in their care during bushfire smoke exposure.

Some caregivers reported their children felt ongoing worry that the smoke will return, often triggered ambiguous environmental cues like overcast skies or parental behaviours they link with their experience of the smoke. This comports with reports elsewhere of the ongoing impact of bushfires on children’s mental health, such as elevated post traumatic stress symptoms and major depression in children at six weeks after experiencing a bushfire [34], and in adolescents three months after their bushfire experience [52]. Building on Wisner et al. [53]’s discussion about communication with children and families after disasters, and Towers [33]’s note that more attention needs to be paid to knowledge and education in the aftermath of disasters, more investigation is needed to establish whether communication principles and needs discussed in the current study apply to recovery once the smoke has dissipated. 

This study is one of the first investigations into the communication needs of children and significant adults in their lives relating to the health impacts of bushfire smoke. The qualitative approach and inclusion of both caregivers and educators is a major strength, as it provides insights from multiple perspectives. However, the relatively small sample size in combination with the wide age range of children discussed limits precision in applying findings to a specific age group. 

## 5. Conclusions

This qualitative exploration of the experiences and communication needs of parents and caregivers of primary-school children has highlighted a number of important considerations for the future development of communication strategies to support children impacted by chronic bushfire smoke exposure. There are both commonalities and differences in children’s experience, which highlight the role of parents and caregivers as models, conduits and curators of information, in particular the need for adults to project calm. We identified air quality apps as the most common source of information used by both adults and children, given the lack of specific child-friendly methods of communication. Language barriers were a major concern for individuals from cultural and linguistically diverse communities. This study has demonstrated the need for the further development of communication strategies designed specifically for children, ideally containing actionable behaviours to proactively protect their health and the health of others, and addressing their focus on the day-to-day experience of bushfire smoke. It also highlights the need for such communications to be available in a variety of languages to serve the community as a whole.

## Figures and Tables

**Figure 1 ijerph-19-12436-f001:**
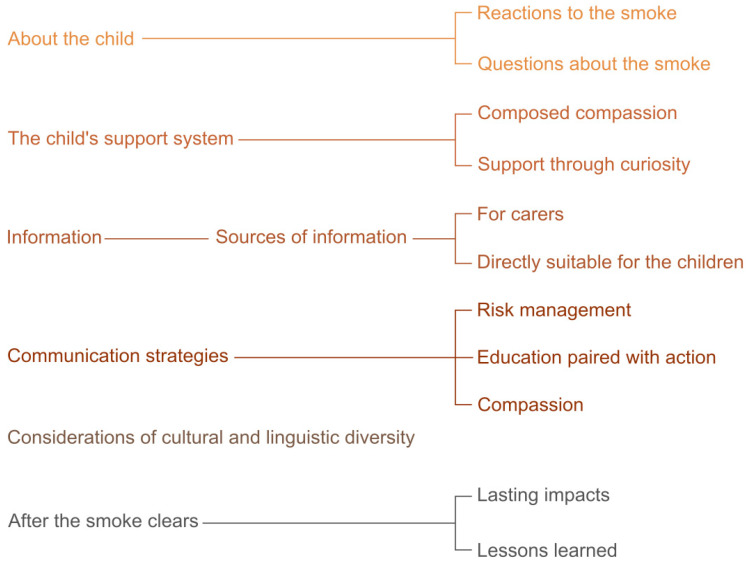
Inductive qualitative conceptual map.

## Data Availability

Not applicable.

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
