# Peer review of "Bushfire Smoke and Children’s Health—Exploring a Communication Gap"

_ijerph, 2022, doi:10.3390/ijerph191912436_

Round 1

Reviewer 1 Report

Dear Editor, 

This study was conducted to provide information regarding the bushfire and its consequences on children's health, having a communication gap. This topic is of great interest and emphasizes the role of parents and caregivers in educating children with diverse ethnic backgrounds. 

However, the primary concern I want to raise is that the study is preliminary and conducted on a small sample size. 

How did the authors communicate with children? Did they hire a translator? 

Can authors provide some more results for additional data? 

Author Response

Reviewer 1: However, the primary concern I want to raise is that the study is preliminary and conducted on a small sample size. 

Response: We note the sample size would be small in the context of quantitative work, however, a sample of this size is often considered adequate for a qualitative paper of this scope. Typically a sample of n=12 is likely to reach saturation, though as few as nine or as many as seventeen interviews may be needed per (Hennink and Kaiser, 2021). In the current study, saturation (that is, the point at which gathering more data about a theoretical construct reveals no new properties, nor yields any further theoretical insights) was achieved around n=12, as the thirteenth interview did not yield any new topics or notably different perspectives on the topics discussed. This said, we do agree that the sample of CALD participants was small and thus findings of this portion should be interpreted with caution; we have added this caveat to page 9, line 393.

------------------------

Hennink, M., & Kaiser, B. N. (2021). Sample sizes for saturation in qualitative research: A systematic review of empirical tests. Social Science & Medicine, 114523.

Reviewer 1: How did the authors communicate with children? Did they hire a translator? 

Response: We did not communicate with children directly. Instead, they spoke with parents, caregivers, and educators who reflected on their experiences with children in the interviews. Two of the authors are multilingual (Burcu Cevik-Compiegne and Karima Laachir) and were able to conduct interviews in languages other than English.

Reviewer 1: Can authors provide some more results for additional data? 

Response: We believe that we have provided a comprehensive analysis of the data and information obtained at the interviews. No additional data are available from these interviews

Reviewer 2 Report

The subject of this review is an article titled Bushfire smoke and children’s health - exploring a communication gap

The text is 13 full pages and 10 lines on page No. 14. The text consists of an abstract, a proper part and a references section. In the proper part, the authors distinguished 5 parts: (1) Introduction, (2) Materials and Methods, (3) Results, (4) Discussion, (5) Conclusions. The most elaborate part of the study is part (3) Results, which consists of as many as six subsections. An additional each of the subsections still has its further parts.

Composition of the article

The reviewed text has a generally balanced composition, but two elements related to its organization raise objections.

The first objection concerns the organization of part (3). "Results" is too disjointed, e.g. number 3.4.1. consists of three lines of text prepared by the authors and two incomplete lines of quotation derived from the research material. Such a strong breakdown of the text makes it difficult to present the research results and does not allow to annotate them. The authors should reconsider the justification for dividing the various parts so finely, taking into account their priorities.

The second objection concerns the "References" section. The main part of the text was on nine full pages (pages 2-10 and on a fragment of page No. 1 - 10 lines of the Introduction). At the same time, the "References" part takes up almost 4 pages. The authors should revise the necessity of such frequent references to the listed publications, evaluating whether each cited source is in fact necessary in the text and has informative significance for the text. The space vacated by the "References" section could be better utilized by presenting the rationale for the study undertaken and the objectives the authors wanted to achieve.

Contents of the article  

The main objections in this review concern the insufficiently precisely defined reasoning for the conducted study, certain elements of its methodology and objectives. The context of the study is objective and thus obvious, but beyond a general description of the "communication gap" the authors do not indicate what it consists of. The text, in the section preceding the study, lacks specification of whether the authors will give equal weight in the study to the cognitive level of communication, the affective level of communication and the volitional level of communication. The objection also relates to the lack of precise indication of whether the analysis was undertaken because of the communication gap associated with communication addressed primitively to adults or addressed primitively to children. If both groups are equally important to the researchers or important to different degrees, the rank should be determined accordingly. As it stands, it is difficult to accurately answer which group is actually the victim of communication gap and for, what reason. If it's adults, the reviewer expects an analysis of communication with adults. If it's children, an analysis of communication with children.

Goals

In the reviewer's opinion, the text lacks a precise specification of the goals, that the researchers want to achieve in the study and about which they want to inform the reader of the article, and how many of these goals there are. As it stands, these intentions must be rather guessed at.  

Methodology

The methodological section, in the reviewer's opinion, lacks some important information for the research procedure.  On page No. 3 there is information that 16 individuals participated in the study. Based on this information alone, it is difficult to assess, whether the research procedure is fruitful and has explanatory value or not. In the reviewer's opinion, it is necessary to introduce information about the size of the community from which the selected group of 16 individuals came. In the description of the procedure, the authors further wrote, that the random sample consisted of 23 people, and the final sample consisted of 16. What made some respondents "drop out" of the procedure? This is important for methodological transparency.

Author Response

Reviewer 2: The first objection concerns the organization of part (3). "Results" is too disjointed, e.g. number 3.4.1. consists of three lines of text prepared by the authors and two incomplete lines of quotation derived from the research material. Such a strong breakdown of the text makes it difficult to present the research results and does not allow to annotate them. The authors should reconsider the justification for dividing the various parts so finely, taking into account their priorities.

Response: The convention used reflects how thematic inductive qualitative content analysis is typically reported, but we acknowledge that this can lead to a somewhat disjointed reading experience. Accordingly, beginning page 4 line 143 we have replaced the more nested subheadings with in-paragraph sentences that can orient the reader.

Reviewer 2:  The second objection concerns the "References" section. The main part of the text was on nine full pages (pages 2-10 and on a fragment of page No. 1 - 10 lines of the Introduction). At the same time, the "References" part takes up almost 4 pages. The authors should revise the necessity of such frequent references to the listed publications, evaluating whether each cited source is in fact necessary in the text and has informative significance for the text. The space vacated by the "References" section could be better utilized by presenting the rationale for the study undertaken and the objectives the authors wanted to achieve.

Response: We are hesitant to remove citations on account of reference list length, especially given the IJERPH specifies no citation number limitations. The manuscript spans multiple disciplines (bushfire smoke health impact, details of the “black summer” smoke plume, child psychology and communications, CALD experience), so many citations only apply once, accounting for the length of the reference list. Revisiting in-text citations, there is little indication of excessive citation for any single point (e.g. a single sentence supported by three or more citations).

Reviewer 2: The main objections in this review concern the insufficiently precisely defined reasoning for the conducted study, certain elements of its methodology and objectives. The context of the study is objective and thus obvious, but beyond a general description of the "communication gap" the authors do not indicate what it consists of. The text, in the section preceding the study, lacks specification of whether the authors will give equal weight in the study to the cognitive level of communication, the affective level of communication and the volitional level of communication.

Response: From the reviewer’s comment, we understand ‘cognitive level of communication’ refers to the child’s capacity to functionally interact with information and extract meaning; ‘affective level of communication’ as the expression of feelings and emotions; and ‘volitional level of communication’ as the degree to which the child choses to communicate. Based on this understanding, we have clarified our originally broad definition of communication and communication gap beginning at page 2, line 99.

Reviewer 2: The objection also relates to the lack of precise indication of whether the analysis was undertaken because of the communication gap associated with communication addressed primitively to adults or addressed primitively to children. If both groups are equally important to the researchers or important to different degrees, the rank should be determined accordingly. As it stands, it is difficult to accurately answer which group is actually the victim of communication gap and for, what reason. If it's adults, the reviewer expects an analysis of communication with adults. If it's children, an analysis of communication with children.

Response: The current study takes a parent/educator/carer perspective of the communication needs of children; we have clarified this in text, page 2, line 98.

Reviewer 2: In the reviewer's opinion, the text lacks a precise specification of the goals, that the researchers want to achieve in the study and about which they want to inform the reader of the article, and how many of these goals there are. As it stands, these intentions must be rather guessed at.  

Response: We have specified our overarching goal at page 2, line 96.

Reviewer 2: The methodological section, in the reviewer's opinion, lacks some important information for the research procedure.  On page No. 3 there is information that 16 individuals participated in the study. Based on this information alone, it is difficult to assess, whether the research procedure is fruitful and has explanatory value or not. In the reviewer's opinion, it is necessary to introduce information about the size of the community from which the selected group of 16 individuals came.

Response: We take this comment to refer to the broader generalizability of our findings, particularly concerns arising from the sample size. In the context of qualitative work, a sample of this size is often considered adequate for a qualitative paper of this scope (typically a sample of n=12 is likely to reach saturation, though as few as nine or as many as seventeen interviews may be needed per Hennink and Kaiser, 2021). We have added the requested estimates of relevant population density at 3, beginning line 108

We cannot estimate specific size of the broader populations for our CALD communities (in particular, the number of relevantly aged children) due to a lack of data from sources such as the Australian Bureau of statistic; and we do acknowledge that generalizability in this subsample may be limited. This consideration has been raised in the discussion (page 9, line 393).

Reviewer 2: In the description of the procedure, the authors further wrote, that the random sample consisted of 23 people, and the final sample consisted of 16. What made some respondents "drop out" of the procedure? This is important for methodological transparency.

Response: We are unable to report on this due to ethical considerations. We are only permitted to disclose information regarding the characteristics, experience, or motivations of individuals who have provided written informed consent, which the seven “drop out” participants did not provide.